# Compassion Fatigue and Perceived Social Support among Polish Nurses

**DOI:** 10.3390/healthcare11050706

**Published:** 2023-02-27

**Authors:** Paulina Pergol-Metko, Anna Staniszewska, Sebastian Metko, Zofia Sienkiewicz, Lukasz Czyzewski

**Affiliations:** 1Department of Development of Nursing and Social & Medical Sciences, Medical University of Warsaw, 02-091 Warsaw, Poland; 2Department of Experimental and Clinical Pharmacology, Medical University of Warsaw, 02-091 Warsaw, Poland; 3Independent Researcher, 00-719 Warsaw, Poland; 4Department of Geriatric Nursing, Medical University of Warsaw, 02-091 Warsaw, Poland

**Keywords:** burnout, compassion fatigue, social support, nursing, Poland

## Abstract

Background: Social support has a vital role in preventing traumatic stress in nurses. Nurses are regularly exposed to contact with violence, suffering, and death. The situation worsened during the pandemic because they were also faced with the possibility of infection SARS-CoV-2 and death from COVID-19. Many nurses are faced with increased pressure, stress, and other adverse effects on their mental health. The study aimed to measure the relationship between compassion fatigue and perceived social support in polish nurses. Methods: The study was conducted on 862 professionally active nurses in Poland using the CAWI method (Computer-Assisted Web Interview). The professional Quality of Life scale (ProQOL) and the Multidimensional Scale of Perceived Social Support (MSPSS) were used for collecting the data. StatSoft, Inc. (2014) was used for data analysis. For comparisons between the groups: Mann–Whitney U test, ANOVA Kruskal–Wallis test, and multiple comparisons (post-hoc). The relationships between variables were tested using Spearman’s rho, Tau Kendall, and the chi-square test. Results: The research showed the presence of compassion satisfaction, compassion fatigue, and burnout in the group of Polish hospital nurses. A higher level of perceived social support was associated with lower compassion fatigue (r = −0.35; *p* < 0.001). A higher level of social support was associated with higher job satisfaction (r = 0.40; *p* < 0.001). The study also found that a higher level of social support was associated with a lower risk of burnout (r = −0.41; *p* < 0.001). Conclusions: Preventing compassion fatigue and burnout should be a priority for healthcare managers. Notably, an essential predictor of compassion fatigue is that Polish nurses often work overtime. It is necessary to pay more attention to the crucial role of social support in preventing compassion fatigue and burnout.

## 1. Background

Compassion is a principal and necessary element in providing professional patient care. It is fundamental for better care and patient outcomes, but several elements can compromise it during healthcare [1]. Healthcare professionals such as nurses are regularly exposed to contact with violence, suffering, and death. In addition to the caregiver role, nurses must provide psychological support for the patient’s family. Chronic stress and anxiety can lead to somatic symptoms and harm the employee’s mental sphere, leading to depression and social problems [2,3].

Stamm (2010) noted that the work of a helper, including work in health care, may affect the quality of professional life. He identified positive and negative aspects of work. He described the positive element of compassion satisfaction (CS) as feeling the pleasure of helping and caring for others and knowing how the work performed can positively impact society. Feelings can enhance nurses’ job satisfaction that they are successful and are well suited to the profession as well as factors such as effective communication and being respected, appreciated, and rewarded. Compassion satisfaction is an emotional response to the rating of self-work life, which is usually addressed both with the concept of motivation. Motivation can impact the behaviours and productivity of nurses responsible for patient life and their professional job satisfaction [4]. The consequence of long-term emotional involvement in caring for people experiencing traumatic stress and suffering, which means the negative aspect of help, is compassion fatigue (CF). Compassion fatigue is the progressive outcome of lengthened and permanent contact with patients, death, and exposure to stress leading to a compassion burden that exceeds nurses’ tolerability levels. Compassion fatigue is a condition where the compassionate energy is on a higher level beyond regeneration, leading to physical, social, emotional, spiritual, and intellectual consequences [5].

Compassion fatigue consists of two parts: the first one is burnout, associated with exhaustion and negative emotions accompanying anger and frustration. Burnout is often described as behaviours of workers who have negative attitudes to work in response to job strain with feelings of disappointment and powerlessness. Burnout among nurses is a significant and prevalent health issue causing severe negative consequences for nurses, patients, colleagues, and health care centres [6]. The second part is secondary traumatic stress (STS) is the fear of doing work and traumatic experiences. Secondary traumatic stress has been defined as feelings and behaviours caused by a traumatic situation experienced by a person. Healthcare professionals are at risk for secondary traumatic stress because they have direct contact with traumatised or suffering patients due to the nature of their work. Frontline healthcare professionals had a higher prevalence of secondary traumatic stress than those working in other hospital units during the COVID-19 outbreak [7]. Studies have shown a strong relationship between the occupational burnout of healthcare professionals and patient safety and medical malpractice. Menon NK, Shanafelt TD, et al. showed an association between depression and suicidal ideation in physicians. Common factors related to suicide or suicidal intentions in the general population include depression, hopelessness, and previous suicide attempts. Burnout is associated with depression, severe mental disorders, and suicidal intentions [8,9]. Permanent exposure to stress and traumatic experiences in the nursing profession significantly contributes to the growth of reduced job satisfaction, burnout, and compassion fatigue, causing a considerably high lack of nurses all around the world [10].

Appropriate management of human resources and improving healthcare professionals’ working conditions could improve employees’ well-being. However, it is worth paying attention not only to economic factors. More and more researchers are looking for psychosocial resources, especially social support, which helps reduce the psychological burden of doctors, nurses, and caregivers [11,12,13,14]. Chen et al. (2017) define social support as an “individual’s experience of being cared for and loved, having a sense of being valued and needed by other people, and being part of a mutually supportive network”. Social support is associated with feeling comfort, respect, happiness, or help from others or a group. Studies show that moderate the adverse effects of physical and mental health burdens [15,16]. Due to the specific work, healthcare professionals cannot avoid stressors, but those who receive social support are less susceptible to compassion fatigue and handle stress better [17].

A sense of coherence and respect helps maintain nurses’ mental well-being [18]. Therefore, social support strongly correlates with burnout and should be pivotal in preventing compassion fatigue and burnout [19,20].

In December 2019, Wuhan, China, diagnosed the first cases of SARS-CoV-2 (severe acute respiratory syndrome coronavirus 2 infection). COVID-19 is caused by SARS-CoV-2, whose most common symptoms are fever, dry cough, and shortness of breath. Most patients also experience other signs of sore throat, headache, myalgia, fatigue, and diarrhoea. Coronavirus is contagious in humans and has rapidly spread worldwide [21].

Moreover, special attention should also be paid to the study date, as it was the beginning of the COVID-19 pandemic. However, the study was designed before the start of the pandemic; that is why the survey did not include questions about feelings and experiences related to the epidemiological situation. Therefore, the obtained results may be related to the situation that prevailed in the world. Since the pandemic, healthcare professionals have reported more physical violence and verbal abuse [22]. Furthermore, nurses’ exposure to mobbing during the pandemic increased. Nurses experienced high levels of burnout during the COVID-19 pandemic, while several sociodemographic, social, and occupational factors. Nurses experience violence at the workplace in the character of every act or threat of verbal or physical force and violence, harassment, intimidation, or other dangerous, threatening, disruptive behaviour that take place at the workplace intending to abuse or injure them.

Nurses around the world every day are working under huge pressure to provide care to suffering and dying patients. The situation worsened during the pandemic because they were also faced with the possibility of infection SARS-CoV-2 and death from COVID-19. Many nurses are faced with increased pressure, stress, and other adverse effects on their mental health [23].

The present study aimed to determine the perception of social support among nurses and identify the relationship between the impact of social support on compassion fatigue and burnout. The above data can help the health manager and psychologists provide social support to nurses to improve their mental health, which can help improve working conditions and prevent burnout.

## 2. Methods

It is an observational, retrospective cohort study conducted using the CAWI method (Computer-Assisted Web Interview) from December 2019–May 2020. The study group consisted of 862 professionally active nurses in Poland. The confidence level is 95; the maximum error is 3%. These nurses volunteered to participate in the study. Six participants were removed from the data analysis due to errors in completing the form. Inclusion criteria for the study: (1) consent to participate in the study; (2) professionally active nurses; (3) work in the hospital.

Stat Soft, Inc. (2014) was used for data analysis. STATISTICA (data analysis software system)—version 12 and Microsoft Excel (version 2016)—Microsoft Office. For comparisons between the groups: Mann–Whitney U test, ANOVA Kruskal–Wallis test, multiple comparisons (post-hoc). The relationships between variables were tested using Spearman’s rho, Tau Kendall, and chi-square tests. The level of significance in all calculations was assumed *p* < 0.05.

The analysis included factors affecting compassion fatigue, compassion satisfaction, and nurses’ burnout. To avoid biases in the study, the authors used standardised scales that had already been used to collect information online. The Likert scale used in the questionnaires allowed the participants to give an answer that did not force a positive or negative response if the respondent’s position was neutral. The survey was supervised, and in case of doubts among the respondents, questions were consulted and answers were given.

The study used the following tools for collecting the data:

### 2.1. Sociodemographic Variables

Sociodemographic data were used to collect information on sex, age, job experience, type of work department, and respondents’ time off work.

### 2.2. The Professional Quality of Life Scale (ProQOL)—Version 5

The professional Quality of Life scale (ProQOL) scale was developed by Dr. Beth Hundall Stamm (1992). The study used the version 5 tool from 2009. The validated 30-element questionnaire includes three 10-element subscales: compassion satisfaction (CS), compassion fatigue (CF), and burnout (BO). ProQOL involves assessing Likert scale answers from 1 to 5, where 1 means “never”, and 5 “very often”. The use of the survey is free and available in various languages. The following study uses a Polish version of the scale [24].

### 2.3. The Multidimensional Scale of Perceived Social Support (MSPSS)

The Multidimensional Scale of Perceived Social Support (MSPSS) author is Gregory Zimet et al. (1988). This instrument consists of three primary sources of support: family, friends, and significant other. The range contains 12 items, rated on a 7-point Likert scale, where 1 means “Very Strongly Disagree”, and 7 means “Very Strongly Agree”. The following study uses Polish adaptation, whose authors were Buszman and Przybyła-Basista (2017) [25,26,27]. An agreement from the author of the Polish adaptation to use the tool in carrying out the study was obtained.

## 3. Results

Finally, 856 respondents were included, consisting of 828 women (96.7%) and 28 men (3.3%). The age of the respondents ranged from 20 to 68. The average for the study group was 39 years old (± SD 11.51). The years of professional experience of the respondents was a maximum of 51. The average for the respondents was 17 years (±SD 12.57). Most respondents took up work in the adult ward (87%). The respondents’ work time was mainly full-time (77%), and 23% were employed full-time plus overtime.

The study showed no correlation between age and compassion fatigue (r = 0.12; *p* = 0.004). There is no relationship between the age of respondents and burnout (r = 0.05; *p* = 0.18) and the level of perceived social support (r = −0.08; *p* = 0.02).

The study showed a relationship between compassion satisfaction and compassion fatigue—the higher the compassion satisfaction, the lower the compassion fatigue (r = −0.45; *p* < 0.001). The higher level of compassion satisfaction was associated with a lower level of burnout (r = −0.71; *p* < 0.001).

The study observed a correlation between the general perception of social support and compassion fatigue (r = −0.35; *p* < 0.001). The obtained results allowed us to identify a significant relationship between compassion fatigue for support from a significant other (r = −0.29; *p* < 0.001), family (r = −0.30; *p* < 0.001), and friends (r = −0.33; *p* < 0.001). A higher level of perceived social support was associated with lower compassion fatigue.

There is a significant relationship between the level of social support and the result of compassion satisfaction. A higher level of social support was associated with greater compassion satisfaction (r = 0.40; *p* < 0.001). The higher level of social support was also associated with a lower risk of burnout (r = −0.41; *p* < 0.001) (Table 1).

An analysis of burnout occurrence was conducted, considering the specificity of hospital wards where Polish nurses work. Due to the possibility of significant differences in results, departments were divided into two particularities: adult and paediatric wards. The types of departments where nurses work was identified, such as ICUs, psychiatric, surgical, and general. In response, “other” respondents often indicated the “A&E department”.

ANOVA analysis indicated that the highest result was obtained by people working in the “surgery” and “other” departments, while the lowest was obtained by those working in the psychiatric ward. The differences in results depending on the department were not statistically significant. (Table 2).

The effect of working time on compassion fatigue incidence among nurses was examined. The study group was divided into two parts: nurses whose working time equals 1 or less and a group working overtime. A job in Poland consists of about 40 h a week—about 165 h per month. Any number of hours over 165 will be overtime. The responses showed that overtime was usually the equivalent of half-time and 1/4 time of employment. The research results suggest that overtime hours are estimated to be between 40 to 82 h a month.

The result of the ANOVA analysis showed significant differences in compassion fatigue depending on working time—people working overtime had significantly higher compassion fatigue than those working full-time or less (Table 3).

## 4. Discussion

This study showed the presence of compassion satisfaction, compassion fatigue, and burnout in the study group. The obtained data indicate that Polish nurses were characterised mainly by average compassion satisfaction, compassion fatigue, and burnout levels. Social support is significant in a profession at high risk of mental and social problems.

Available studies show that psychosocial risks are related to health problems, work accidents, low job satisfaction, burnout, and work-related stress. All these factors may predispose to a cardiovascular burden and social isolation. These people have a higher incidence of mental illness, such as depression and anxiety. Properly managing psychosocial risks and responsibilities helps increase productivity, prevent malpractices, and improve well-being [28].

The results of the study indicated no relationship between the age of respondents and compassion fatigue. Other studies regarding age indicate that higher age is associated with a higher compassion satisfaction level [29]. Similar to the Zhang study, the results show that the positive aspects of work, like compassion satisfaction, correlate with a lower intensity of burnout [30]. These results may indicate that job satisfaction positively affects nurses’ mental health and well-being [31]. According to Cheung and Lee, a significant problem in achieving job satisfaction is the aggressive behaviour experienced by healthcare professionals. Of nurses with low job satisfaction, 79.7% reported a fear of violence, which was associated with the experience of bullying and physical attack in the past [32]. Nurses with a higher level of perceived social support had lower levels of compassion fatigue and burnout. Similar results were obtained in the study of Iranian nurses. Nurses whose perceived social support score was higher were characterised by a lower level of burnout and, in the case of family support, a lower level of compassion fatigue [33]. Other studies proved that higher compassion satisfaction was shown by nurses who receive social support [34].

Many hospital nurses in Europe work 12-h shifts and overtime. Even though nurses do not complain about such a high number of hours of work per day, research shows that working over a standard 8 h of work per day is associated with greater dissatisfaction with work (aOR = 1.40; 95% CI 1.20 to 1.62) and a greater intent to leave (aOR = 1.29; 95% CI 1.12 to 1.48) [31]. The results of our study are, therefore, coherent with the results of the Luther and Gerhart study in which overtime-working respondents experienced burnout in three indicators: emotional exhaustion and lack of personal achievement [35].

Khatatbeh H et al. investigated the relationships of family, co-workers, and manager support with paediatric nurses’ satisfaction and their perceived patient adverse events. Conducted study shows that older nurses perceive less frequent medication errors and pressure ulcers. This disclosure might be related to the more experience the nurses will have and the more knowledge of patient safety significance. Nurses’ job satisfaction causes less frequent patient adverse events like pressure ulcers [36]. During the pandemic, paediatric nurses reported that they experienced high emotional exhaustion, low personal accomplishment, and moderate compassion fatigue. Burnout may affect the quality of care nurses provide, but it could also negatively affect their well-being and quality of life [37]. During the COVID-19 pandemic, nurses experienced a psychological crisis shown by Li X and Wang H. Better social support involves fewer negative emotions, including depression or anxiety [38]. Similar results were obtained in the study by Hu D, Kong Y, Li W, et al., which showed that 40% to 45% of the frontline nurses experienced anxiety or depression, with 11% to 14% having moderate to severe anxiety or depression [39]. Another study showed that social support could moderate the effect of job strain on organizational commitment and burnout. Li N, Zhang L, et al.’s findings demonstrated that emotional exhaustion to depersonalization was different between the low and high social support groups. Nurses overburdened with work and low social support were more likely to have depersonalization. The same study revealed that nurses under job strain could have emotional exhaustion, which could lead to mental disorders like depersonalization and may influence their work. Furthermore, emotional exhaustion and depersonalization had a crucial role between job strain and commitment to work. That shows how social support involves the relationships among workload, emotional exhaustion, depersonalization, and commitment to work [40]. Our study showed that some nurses work overtime and, as in the study by Bae SH, Pen M, Sinn C, et al., it can be assumed that this is one of the reasons for the fatigue and burnout of nurses. Overtime work was mainly related to the need to earn extra money or due to a lack of nursing staff [41]. Burmeister EA, Kalisch BJ, et al. identified an essential difference between countries with absenteeism and intent to leave in nurses. Increased lack of staffing, lower job satisfaction, little experience of nurses, and younger age were significant contributors to nurse absenteeism and intent to leave. Results indicated that introducing psychological interventions like mindfulness and cognitive behavioural therapy is an effective method for anxiety, reducing stress and depression. Additionally, teaching relaxation techniques such as deep breathing may be beneficial. This study also points to the significant role of physical activity in well-being. It is recommended to use pedometers, and health coaching with texting may help to increase physical activity. [42].

Healthcare managers should promote the practice of self-care by engaging in proper nutrition and encouraging work-life balance, either to save their health or to be role models for other nurses in direct patient care. As shown in a study by Ross A, Yang L, et al., nurses who enjoy their jobs could experience less anxiety and stress. They also may have more energy to do training and to prepare and consume healthy meals [43]. Moreover, some nurses described increased alcohol consumption caused by the COVID-19 pandemic, mostly due to the stress of working in an environment where there was no personal protective equipment. Nurses were also burdened with overtime, lack of breaks, and increased workload. Increased alcohol consumption was associated with burnout, absenteeism, and intention to leave. The nursing profession is currently undergoing significant continuing stress in providing care and management to patients with the SARS-CoV-2 virus, and increased alcohol consumption is a significant threat to personal and workforce well-being, workforce sustainability, and quality nursing care [44]. There was a global shortage of nurses worldwide, but it may be made worse by the increased demands of caring during COVID-19 and the usual care of non-COVID patients. However, there is currently no data on differences between staff shortages before and after the acute phase of the pandemic. One significant apprehension is that the pandemic and its numerous effects on the nursing profession will worsen fatigue in nursing attrition and poor mental health in the future, which may make them intend to leave the profession. Another serious question is if the profession will attract a sufficient number of newly educated nurses to care for the population in the future. Governments and health managers around the whole world should invest in health care and nursing and pay more attention to the needs of health systems to provide a healthy population. It is argued that without this, economies will not get well and prosper, and medical systems will not be able to provide quality care [45].

There are some limitations to this study. First, it would have been necessary to investigate why many such nurses work overtime because the motivation could be the respondents’ social, family, or economic situation, not just a low salary. It should be necessary to perform psychological tests among the study participants regarding psychological burdens related to work and private life because private life also impacts the development of burnout. Studies showed that family–work conflict appeared to increase levels of perceived psychological effort [46]. An additional advantage of the study could have been to divide the respondents into the private and public sectors due to differences in earnings and working conditions.

## 5. Conclusions

Compassion fatigue and burnout in nurses are severe problems in the healthcare system. It is worth noting that an essential predictor of compassion fatigue is that Polish nurses often work overtime. The employer should provide psychological care to enhance emotional intelligence in the workplace. It helps prevent adverse effects of psychosocial risks such as burnout and psychosomatic complaints and positively influences job satisfaction. Healthcare managers should pay special attention to the number of working hours and the mental health of nurses and other healthcare workers. Attention should also be paid to the crucial role of social support in preventing burnout of healthcare providers. Effective prevention of compassion fatigue and burnout could improve work efficiency, reduce absenteeism at work, and reduce medical errors. It is possible thanks to a policy aimed at improving the working conditions of all healthcare professionals. The implementation of well-being interventions has been proven in research. For example, Dincer and Inangil implemented a program of emotional freedom techniques that decreased burnout in healthcare professionals. Lee et al. promoted the coping strategies program to improve lower burnout levels among healthcare personnel. Both studies observed positive correlations between burnout and secondary trauma or compassion fatigue [47,48,49].

In conclusion, the mental strain on healthcare workers should be a priority for people involved in human resources management. Further qualitative research is needed to identify compassion fatigue and burnout determinants among nurses. Healthcare managers can improve nurses’ quality of life by increasing social support and strengthening psychological resilience. They should pay attention to the support provided to nurses and take proactive measures to increase self-esteem to effectively manage stress and negative emotions from work and life to raise their quality of life.

## Figures and Tables

**Table 1 healthcare-11-00706-t001:** Results of Simple Regression Analyses.

Characteristic	Compassion Fatigue		Burnout		Compassion Satisfaction		Social Support	
	r	*p*	r	*p*	r	*p*	r	*p*
**Age**	0.12	0.004	0.05	0.18	0.03	0.36	−0.08	0.02
**Compassion Satisfaction**	−0.45	<0.001	−0.71	<0.001	-	-	0.40	<0.001
**Social Support**			0.41	<0.001	0.40	<0.001		
Friends	−0.35	<0.001					-	-
Family	0.33	<0.001					-	-
Significant	0.30	<0.001					-	-
Others	0.29	<0.001					-	-

**Table 2 healthcare-11-00706-t002:** The level of burnout by the particularity and type of work department.

	Particularity	Type	N	Mean	Mdn.	Min.	Max.	Q1	Q3	±SD	*p*
**BO**	Pediatric	ICU	28	24.29	24	16	34	21	26	4.78	0.417
		Psychiatric	1	25.00	25	25	25	25	25	-	
		Surgery	19	26.63	26	17	43	20	31	6.82	
		General	27	23.89	24	11	34	20	28	5.51	
		Other	35	25.14	25	13	37	20	31	6.81	
	Adult	ICU	127	25.34	25	11	38	22	29	5.44	
		Psychiatric	41	23.83	25	12	36	18	28	6.16	
		Surgery	218	25.15	25	14	38	21	29	5.49	
		General	171	25.77	26	13	41	22	30	5.91	
		Other	189	25.36	25	10	42	21	29	6.07	

**Table 3 healthcare-11-00706-t003:** Impact of working time on compassion fatigue.

	Working Time	N	Mean	Mdn.	Min.	Max.	Q1	Q3	±SD	*p*
**CF**	1 or less	659	51.80	51	22	85	45	59	10.73	0.043
	overtime	197	53.40	54	28	78	46	61	10.89	

## Data Availability

The datasets used and analyzed during the current study are available from the corresponding author upon reasonable request.

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
