# Peer review of "Compassion Fatigue and Perceived Social Support among Polish Nurses"

_healthcare, 2023, doi:10.3390/healthcare11050706_

Round 1

Reviewer 1 Report

First of all, congratulations to the authors for their work.

In relation to the introduction I think it is well contextualised, I would recommend you to add the following articles.

Sousa, L., Ferreira, B., Silva, P., Tomás, M., José, H., Garcia-Navarro, E. B., & Ortega-Galán, Á. (2022). Bibliometric analysis of the scientific production on compassion fatigue. Journal of Personalized Medicine, 12(10), 1574-https://doi.org/10.3390/jpm12101574.

Ruiz-Fernández, M. D., Alarcón-Ortega, C., Ventura-Miranda, M. I., Ortega-Galán, Á. M., Alcaráz-Córdoba, A., Berenguel-Marínez, A., & Lirola-Manzano, M. J. (2022). Burnout in specialized care nurses during the first COVID-19 outbreak in spain.10(7), 1282-https://doi.org/10.3390/healthcare10071282.

Regarding the methodology, I think it is very up-to-date to carry out the surveys using the CAWI method.

In the Results section, in the title of table 1, I would not put abbreviations (CS, SS and CF, BO, CS, SS).

The improvement I would propose to this work would be to carry out the same study on the same population but asking them about their work during the pandemic by COVID-19 and to carry out a comparative study to see if there has been an increase in compassion fatigue and burnout during work in times of pandemic.

Author Response

Dear Reviewer,

Thank you for all of your comments.

We removed abbreviations from the title of the table.

I think conducting a study among the same population after the COVID pandemic is an interesting idea, and maybe we will use it for other research.

Kind Regards

Paulina Pergol- Metko

Reviewer 2 Report

Compassion fatique as a research category showing the effects of physical, psychological and emotional help to others (stress, trauma, etc.) is becoming more and more important theoretically and practically nowadays. It is not only the essence of some professions (e.g. nurses, social care, rescuers, doctors, etc.), but the situation of performing face-to-face human service activities, such as salesmen, teachers, prison employees, etc., etc. Aggressive behavior is mentioned here other stakeholders, counterproductive employee behavior, intimidation, etc.

The conducted research concerns the verified opinions of 856 people from a group of 862 hospital nurses, and was carried out in the pre-pandemic period (no reference to the COVID-19 period). The authors adopted implementation methods and tools adequate to the assumptions. They used research tests recognized in the literature, such as the ProQOL and MSPSS scales, and the CAWI method with the consent of the study participants. Calculations were performed using STATISTICA. The studies are technically correct and include the necessary methodological calculations between groups (Mann-Whitney U test, ANOVA Kruskal Wallis test, multiple comparisons) and between variables, such as Spearman's R test, Kendall's Tau test and chi 2 test. Conclusions are factually correct and consistent with scope of research.

The article contains theoretical and practical conclusions from the research. The research results show a theoretically significant problem of the multi-level occurrence of this category in the hospital environment, i.e. the differences between people working in different wards. Differences are also due to different workloads (overtime). The 12-hour working day is also a negative factor. This is compounded by significant practical problems, such as, for example, shortages of nurses, the "aging" of this professional population, also due to lower interest in the profession, or increased temporary personnel burdens (e.g. during the COVID-19 period). This creates big problems in the organization of health care work. The methods of support proposed by the authors are just the beginning of creating theoretical and practical methods of operation in occupations defined in the organizational literature as high-risk occupations (e.g. Johns, 2010, Some unintended consequences of job design, Journal Of Organizational Behaviour, 31(2-3 ), 361-369). This is due to their social necessity

For the above reasons, I recommend publishing the article

Author Response

Dear Reviewer,

Thank You for your valuable opinion.

Kind Regards

Paulina Pergol- Metko

Reviewer 3 Report

This paper examines the association between perceived social support and compassion fatigue (CF) in a sample (n = 862) of Polish nurses. As previous review articles have identified important lacunae in the existing literature on CF in nursing staff, the current paper has value as an incremental advance or contribution to this literature.

There are certain aspects of the paper that would benefit from correction or clarification:

1. In the Introduction (line 50 onwards), the authors mention that CF has two components - burnout and secondary traumatic stress. This is not a universally accepted concept. Some authors refer to CF as being distinct from burnout in certain key aspects. The authors should provide a reference for their description of the CF construct, and briefly discuss contrasting views.

2. The discussion of depression and suicide in lines 62-66 seems out of place; it would make more sense to highlight this issue after mentioning the available evidence on the links between CF and depression. (It is also worth noting that these links may be bidirectional: CF may increase the risk of depression, but an individual suffering from depression may experience cognitions similar to those seen in CF.)

3. In the three paragraphs beginning at line 86 onwards, the authors have chosen to highlight the impact of the COVID-19 pandemic on CF, social support, and mental health among nursing staff. If this is one of the key aspects of the study, it could be mentioned in the title and abstract as well. On the other hand, if the study data was collected prior to the impact of the COVID-19 pandemic in Poland, the paragraphs related to COVID-19 could be shortened and shifted to the Discussion section.

4. In line 117 (Methodology), what is meant by "trust level"? This should probably read "confidence level". Alternately, the authors could use the technical terms alpha and beta. Additional details on the estimation of the study's sample size would also be helpful.

5. Data analysis should be described in a separate paragraph after the description of the study tools. In this section, besides listing the tests that were used, the rationale for the use of these tests should be clearly stated (e.g., if the authors have used non-parametric tests, they should clearly mention that the data did not conform to a Gaussian distribution.) Also, the names of some of the tests require minor correction (e.g., Spearman's rho instead of "Spearman's R"; chi-square instead of "chi 2").

6. Details on the demographic / work-related characteristics of the study participants (lines 122-127) could be shifted to the Results section.

7. When formatting the study results / tables, please check the journal's style recommendations. The authors have used the , (comma) instead of the decimal point in several places.

8. Multiple bivariate correlation analyses have been presented in Table 1. Were these corrected for multiple comparisons, to minimize the risk of a false-positive finding? If not, why?

9. In Table 3, working hours have been divided into "1 or less" and "overtime". Would it be more informative to examine bivariate correlations between the number of working hours (as a continuous variable) and measures of CF?

10. Did the authors test whether social support mediated / moderated the associations between other risk factors (e.g., age and working hours) and CF? This would be an interesting finding to report.

11. Language editing is required to address several minor errors in spelling ("family" instead of "familly"), word choice ("particularity" in Table 2 could be replaced by a simpler word) and sentence structure.

Author Response

Dear Reviewer,

Thank You for your revision. 

We tried to improve the manuscrip according to your comments.

  1. We have covered each aspect of compassion fatigue. We do not focus on burnout in general but on what it means specifically from the perspective of the study.
  2. We wanted to strongly emphasize the real dangers of compassion fatigue, burnout and traumatic stress.
  3. data collection coincided with the beginning of the pandemic, therefore we do not have data from the period before the pandemic. The effects of the pandemic were described in the discussion
  4. It have been corrected
  5. It have been corrected
  6. It have been corrected
  7. It have been corrected
  8. Yes, it was corrected for multiple comparisons
  9. probably it would be, but we didn't study the specific amount of overtime and it could be difficult to measure it without those data. We will be happy to use your feedback in the next study
  10. We haven't tested it but thanks for the suggestion. We will gladly use it in the next study
  11. We've tried to improve linguistic errors like this

As suggested, we changed the abstract and corrected linguistic errors in the methodology and manuscript. We plan to conduct another study after the acute phase of the pandemic is over, and then we will be happy to include all suggested corrections and threads in the new research.

Kind Regards

Paulina Pergol- Metko